# Secondary Metabolites and Their Antioxidant Activity Enhance the Tolerance to Water Deficit on Clover *Lotus corniculatus* L. through Different Seasonal Times

Luis Angel González-Espíndola [1], Aurelio Pedroza-Sandoval [1,*], Gabino García de los Santos [2], Ricardo Trejo-Calzada [1], Perpetuo Álvarez-Vázquez [3] and Maria del Rosario Jacobo-Salcedo [1]

[1] Unidad Regional Universitaria de Zonas Aridas de la Universidad Autonoma Chapingo, Km. 40 Carretera Gómez Palacio-Chihuahua, Bermejillo C.P. 35230, Durango, Mexico; qfbgonzalez_espindola@hotmail.com (L.A.G.-E.); rtrejo@chapingo.uruza.edu.mx (R.T.-C.); jacobo.rosario@inifap.gob.mx (M.d.R.J.-S.)

[2] Colegio de Postgraduados, Campus Montecillo, Km 36.5 Carretera Mexico-Texcoco, Montecillo, Texcoco C.P. 56230, Estado de México, Mexico; garciag@colpos.mx

[3] Departamento de Recursos Naturales Renovables de la, Universidad Autónoma Agraria Antonio Narro, Calzada Antonio Narro 1923, Buenavista, Saltillo C.P. 25315, Coahuila, Mexico; perpetuo.alvarezv@uaaan.edu.mx

\* Correspondence: apedroza@chapingo.uruza.edu.mx; Tel.: +52-871-887-3275

**Abstract:** This study aimed to evaluate the effects of a water limitation in different ecotypes and one variety of *Lotus corniculatus* L. on the production of secondary metabolites and their antioxidant activity in response to a water deficit (WD) through other seasonal times. A randomized block experimental design with three replicates was used. Two levels of soil water content and five genotypes were arranged in a factorial way ($2 \times 5$) with ten treatments for replication. The 255301 ecotype showed significantly higher ($p \leq 0.05$) concentrations of total phenols, with a concentration of 86.6 mg Gallic Acid Equivalent (GAE)/gram of fresh weight (gFW); total flavonoids, with a concentration of 63.2 mg Quercetin Equivalent (QE)/gFW; total tannins (71.7 mg GAE/gFW); and radical scavenging activity, with an average of 200 mg Trolox Equivalent Antioxidant Capacity (TEAC)/gFW in winter under a WD. The 255305 ecotype showed an increase in radical scavenging activity of 230 mg (TEAC)/gFW and a total tannin concentration of 65.3 mg GAE/gFW in winter and spring, respectively, under a WD. The 255301 ecotype showed an increase in the concentration of total saponins (254.8 mg saponins/gFW) in summer under a WD. All these responses were triggered to mitigate a water deficit and extreme temperatures.

**Keywords:** stress physiology; biochemical processes; drought; forage crops; soil moisture

## 1. Introduction

Climate change has modified weather patterns with increasingly extreme events. Rainfall is one of the climatic components with an environmental impact causing extreme events in the world [1]. In addition, the world's rising population, now pegged at more than 8 billion, is putting intense pressure on using natural resources for agri-food production [2]. Livestock is a supplier of agri-food products and an essential protein source in the family diet. Consumers' high demand for meat, milk, and other derivatives has increased animal production and led to a greater need for forage [3]. Forage production consumes 76% of the freshwater available for productive activities on the planet [4], which is a significant problem for regions due to the scarcity of this natural resource.

Drought and intensive agri-food production activity cause environmental deterioration in arid areas [5] where the productivity is with a high-water demand. This makes it necessary to perform strategies to produce forage through alternative crops with a lower water consumption and the ability to compete with other crops in quality and quantity [6,7].

Alfalfa (*Medicago sativa* L.) is the main forage crop, with the best nutritional quality and quantity of fresh and dry biomass produced as food for livestock. Paradoxically, however, it is one of the crops with the highest water consumption [8].

Several species are studied as alternative forage crops in the leading livestock areas of the world, with studies related to the evaluation of water efficiency, environmental adaptation, and productivity. Among these species are white clover (*Trifolium repens* L.), red clover (*T. pretense* L.), pink clover (*T. fragiferum*), and birdsfoot treefoil (*Lotus corniculatus* Lam.) [9]. The *Lotus* genus has more than two hundred annual and perennial species with different growth habits, such as prostrate, erect, and semi-erect forms [10]. Most ecotypes and varieties of *L. corniculatus* are grown in regions with a humid temperate climate [11]. Few studies have been conducted under extreme conditions, such as those of arid lands [10]. However, *Lotus* has considerable genetic diversity, which is an opportunity to explore the response capacity of the genetic resources of this genus, which could adapt to conditions of low water availability and extreme temperatures.

The plants have several morphological, physiological, and chemical mechanisms to tolerate extreme environments [12]. Some physiological indicators, such as phytochemicals produced through secondary metabolism, named secondary metabolites, are highly reactive in plants under environmental stress, such as a water deficit, soil salinity, extreme temperatures, and a nutrient deficiency [13,14]. They play a crucial role in the survival of plants in marginal conditions since they are activated in response to environmental stress and function as agents and physiological regulators [15]. Secondary metabolites, such as phenolic compounds, alkaloids, and terpenoids, have been shown to have antioxidant and antimicrobial properties, which help mitigate oxidative damage caused by water scarcity [16,17].

Furthermore, reactive oxygen species (ROS) are increased under water stress as molecular signals that activate adaptive responses in plants [18]. ROS and secondary metabolites' activity interaction is a complex process that allows plants to survive in extreme environments [19]. This study aimed to evaluate different ecotypes and one variety of *Lotus corniculatus* L. in terms of its ability to produce secondary metabolites with antioxidant activity under a water deficit through other seasonal times.

## 2. Materials and Methods

### 2.1. Geographic Location

The experiment was conducted at the experimental field of the Unidad Regional Universitaria de Zonas Aridas of the Universidad Autonoma Chapingo at Bermejillo Mapimí, Durango, Mexico. The region is located at 25.8° NL and 103.6° WL and an elevation of 1130 m. The area has a desert climate with rain in summer and cool winters, with an average annual rainfall of 258 mm, average potential evaporation of 2000 mm, and an average temperature of 21 °C, with a maximum of 33.7 °C and a minimum of 7.5 °C [20]. Temperatures recorded inside the shade mesh were extremely high in summer, low in winter, and moderate during spring and autumn. The experimental area within the shade mesh was covered in advance with a plastic cover to avoid alterations in the soil moisture content in the pots during rainy periods.

### 2.2. Experimental Setup

A randomized block experimental design with three replicates was used under shade mesh conditions. The treatments of soil water content were as follows: (1) no water deficit (NWD) with 100% of field capacity (FC), and (2) water deficit (WD) with 89% of FC in four ecotypes identified with ID numbers 255301, 255305, 202700, and 226792, and one variety named Estanzuela Ganador of *L. corniculatus*, which was obtained from different areas, such as Europe, North America, and South America, with a temperate humid climate (Table 1).

**Table 1.** Identification, country of origin, and growth habits of different plant genetic resources of *Lotus corniculatus* L.

| Ecotype Identification/Variety * | Country of Origin | Growth Habit |
|---|---|---|
| 255301 | France | Semi-erect |
| 255305 | Italy | Semi-erect |
| 202700 | Uruguay | Erect |
| 226792 | Canada | Semi-erect |
| Estanzuela Ganador | Uruguay | Erect |

\* Donated by the Institute of Natural Resources of the Postgraduate College at Montecillo, Mexico.

The Lotus genetic materials evaluated in this study were selected based on their ability to survive the environmental conditions in the study area, from a total of 12 original genetic materials: 4 varieties and 8 different accessions.

Ten treatments were established of the factorial $2 \times 5$. The experimental unit was a plant in a pot with three replicates. Seedlings were previously grown in 1 kg black plastic bags in soil mixed with compost at a 3:1 ratio for two months. After this time, plants of approximately 20 cm in height were transplanted into rigid 18 kg plastic pots containing a substrate prepared to have a ratio of 50:30:20 of soil, compost, and sand, respectively. According to a physical and chemical analysis of the substrate, it was 26% silt, 22% clay, and 52% sand, with a pH of 7.73, electric conductivity (EC) of 7.47 dS/m, and bulk density of 1.46 g/cm$^3$ (Figure 1) The experiment was carried out from March 2021 to May 2022.

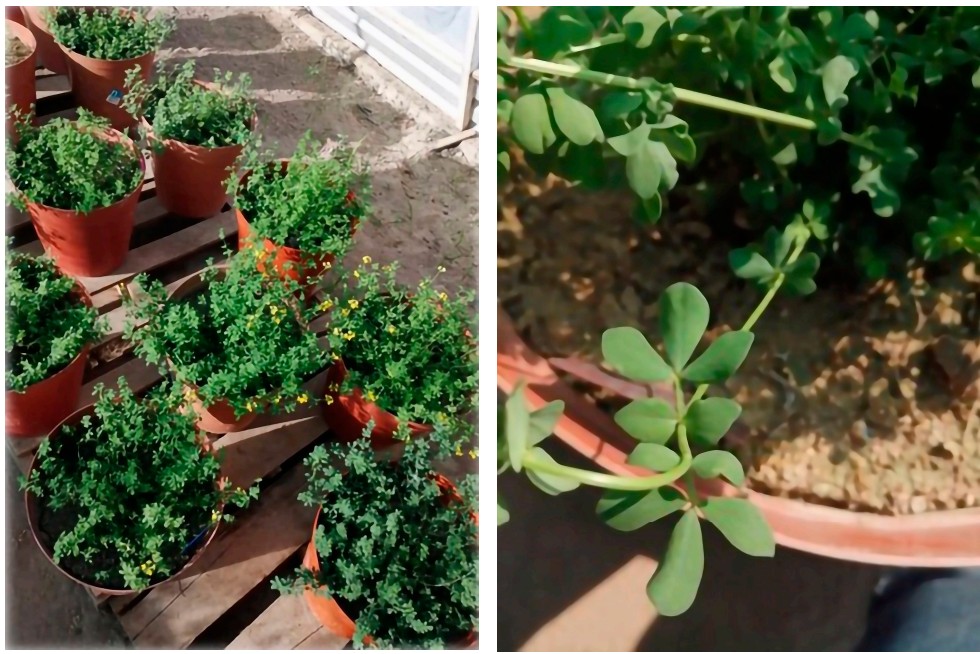

**Figure 1.** A view of part the experiment of *Lotus corniculatus* L. under shade mesh. Bermejillo, Dgo. México.

According to the determination using the membrane pot method [21], the substrate used in the pots, the soil moisture content at the field capacity (FC), and the permanent wilting point (PWP) were 27.5% and 17.5%, respectively (Figure 2).

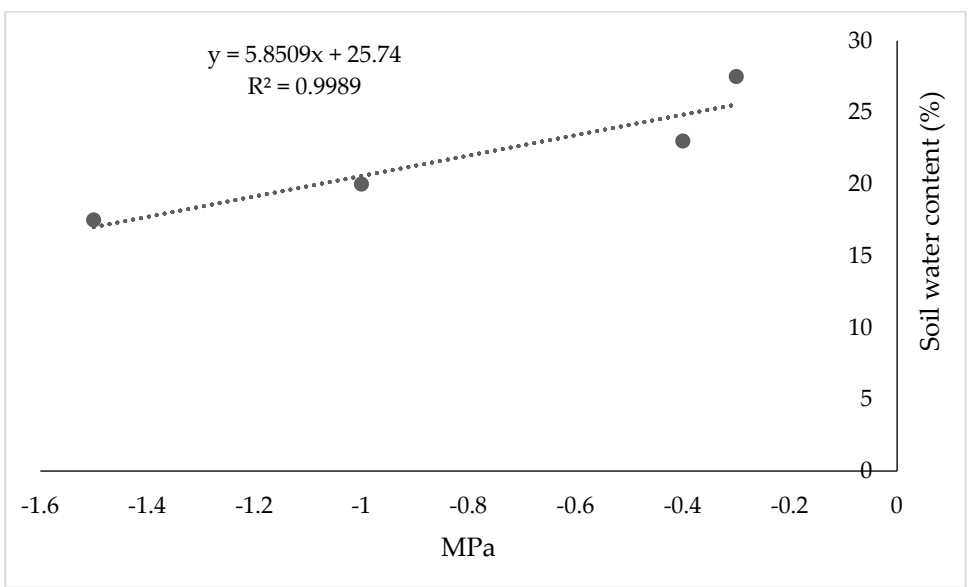

**Figure 2.** The membrane pot method calculates the down curve of soil moisture (10).

*2.3. Irrigation*

Irrigation was performed manually, and during the first 15 days of the experiment, the soil moisture was maintained at FC (27.5%) for all treatments. Subsequently, the water content in the soil was restricted until maintaining the differentiated ranges of 26.5% $\pm$ 1 and 23.5% $\pm$ 1. The upper limit of soil water content in the first range was 27.5%, corresponding to 100% of FC, and the upper limit of the second range was 24.5%, corresponding to 89% of FC. This water decrease in the soil was decided since *L. corniculatus* is a plant with a C3 photosynthetic pathway, which is sensitive to water stress [22].

The different genetic resources of *L. corniculatus* were homogenized in the plant canopy by cutting the biomass at 62 days after transplanting (DAT). Since the plant grows by forming a rhizome with several new shoots, the cuts of fresh matter on the different sampling dates were made with pruning shears 6 cm above soil level, using a plastic ring that allowed the cutting height to be uniform [23]. From this date, seasonal cutting intervals were defined: two cuts at intervals of 42 days in each seasonal period (summer, autumn, and spring) and 92 days in winter (a single cut), the latter interval defined by the slow growth of the plant due to low temperatures.

The soil water content was measured regularly in real-time using a digital tensiometer (Model: MO750, Extech Instruments Co., Laredo, TX, USA). When the soil moisture content reached the lowest limit of a treatment, irrigation was resumed until the upper limit of each irrigation treatment. During the experimental time, the environmental temperature and humidity inside the shade were recorded daily using a digital thermometer-hygrometer (Model OUS-WA62, ORIA, Shanghai, China).

*2.4. Measured Variables*

Fresh tissue of *L. corniculatus* samples was used to quantify secondary metabolites according to the method reported by Melgarejo et al. [24]. All samples were stored at $-40$ °C until they were used. The total phenols concentration (TPC) and total tannins concentration (TTC) were measured using Folin–Cioucalteu reagent [25]; the extract was prepared with 1 mg of plant tissue in 1000 μL of methanol, which was shaken for 1 h and subsequently centrifuged at 4000 rpm for 15 min. These variables were measured in mg Gallic Acid Equivalents per gram of fresh weight (GAE/gFW) [26]. The total flavonoid concentration (TFC) was quantified using the method described by Maksimovíc et al. (2005) [27] with modifications. Quercetin was the standard, so the result was expressed in mg Quercetin Equivalents (QE/gFW). At the same time, total saponins concentration (TSC) was calculated using colorimetry with Lieberman–Burchard reagent [28] in mg

of saponins/GFW. Radical Scavenging Activity (RSA) was measured using absorbance of the DPPH (2,2-diphenyl-1-picrylhydrazyl) radical [29] reported in Trolox Equivalent Antioxidant Capacity (TEAC)/gFW.

*2.5. Data Analysis*

Data were analyzed with normality tests, one-way ANOVA, Tukey's range test, and simple linear regression using the PASW Statistics statistical program for Windows 18.0.0 Chicago, IL, USA, SPSS Inc.

## 3. Results

*3.1. Temperature and Relative Humidity*

The temperature was recorded within the shade mesh from June 2021 to May 2022. An average maximum temperature of 30 °C, an average minimum of 20 °C, a maximum of 46.9 °C, and a minimum of −4.6 °C were recorded. The middle and maximum temperatures per day were 16.6 and 40.1, 19.8 and 32.7, 9.8 and 37, and 4.5 and 30 °C in spring, summer, autumn, and winter, respectively. Along with the experiment's relative humidity, the average was 57.5%, with a minimum mean of 42% and a maximum mean of 57.5% (Figure 3). The experimental area within the shade mesh was covered in advance with a plastic cover to avoid alterations in the soil moisture content in the pots during rainy periods. July and September had the highest rainfall, with 44 and 54 mm averages, respectively.

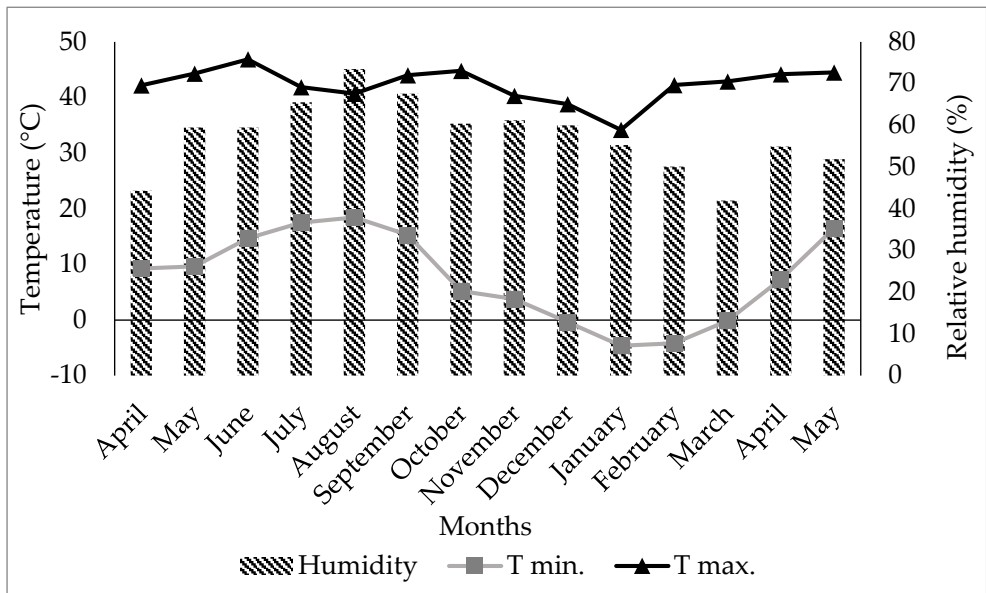

**Figure 3.** The performance of maximum, minimum, and mean temperatures (°C) and mean relative humidity (%) inside the shade mesh from April 2021 to May 2022.

*3.2. Phenols*

The total phenols concentration (TPC) in the tested plant resources was significantly different ($p \leq 0.05$). In winter 2021–2022, the 255301 and 226792 ecotypes showed a higher concentration with values of 86.6 and 87.2 GAE/gFW under a WD. At the same time, the lowest concentration was found in 226792 ecotype under NWD (Table 2).

**Table 2.** Total phenols concentration in different ecotypes and one *L. corniculatus* variety under water deficit throughout the seasons.

| Ecotypes/Variety | SWC | Total Phenols Concentration (mg GAE/gFW) | | | |
|---|---|---|---|---|---|
| | | Summer 2021 | Autumn 2021 | Winter 2021–2022 | Spring 2022 |
| 255301 | 100% FC (NWD) | 38.793 ± 4.5 [a] | 20.134 ± 3.4 [a] | 61.325 ± 4.5 [b] | 25.359 ± 3.2 [a] |
| | 89% FC (WD) | 47.301 ± 0.5 [a] | 33.275 ± 2.6 [a] | 86.622 ± 3.8 [a] | 31.116 ± 1.3 [a] |
| 255305 | 100% FC (NWD) | 50.670 ± 1.5 [a] | 27.374 ± 5.3 [a] | 73.453 ± 1.8 [ab] | 29.323 ± 0.7 [a] |
| | 89% FC (WD) | 45.181 ± 7.7 [a] | 26.850 ± 1.9 [a] | 81.437 ± 5.7 [ab] | 30.906 ± 1.4 [a] |
| 202700 | 100% FC (NWD) | 41.521 ± 8.8 [a] | 20.848 ± 3.4 [a] | 62.495 ± 5.2 [b] | 23.894 ± 3.5 [a] |
| | 89% FC (WD) | 40.451 ± 1.4 [a] | 29.363 ± 2.8 [a] | 76.156 ± 6.2 [ab] | 30.715 ± 1.0 [a] |
| 226792 | 100% FC (NWD) | 38.975 ± 1.9 [a] | 28.591 ± 1.2 [a] | 61.450 ± 0.7 [b] | 25.035 ± 0.2 [a] |
| | 89% FC (WD) | 39.017 ± 3.4 [a] | 31.117 ± 4.9 [a] | 87.248 ± 1.4 [a] | 36.392 ± 3.0 [a] |
| Estanzuela Ganador | 100% FC (NWD) | 49.593 ± 5.4 [a] | 30.404 ± 3.8 [a] | 66.796 ± 2.0 [ab] | 25.353 ± 3.8 [a] |
| | 89% FC (WD) | 38.962 ± 1.9 [a] | 27.781 ± 1.8 [a] | 73.926 ± 6.7 [ab] | 33.561 ± 3.0 [a] |

Mean ± standard deviation. Tukey test ($p \leq 0.05$). Figures with the same letter within the same column are statistically equal. SWC = soil water content; GAE = Gallic Acid Equivalent; FC = field capacity; NWD = no water deficit; WD = water deficit.

*3.3. Flavonoids*

The total flavonoid concentration (TFC) was remarkably similar to the TPC performance with a statistical difference ($p \leq 0.05$). Only in the winter was the Estanzuela Ganador variety significantly higher, with a concentration of 63.2 mg QE/gFW in a WD. In comparison, the 226792 ecotype had the lowest value with 27.1 mg QE/gfw under NWD. The TFC was quantified as 12.8–63.2 mg QE/gFW within the evaluated time (Table 3).

**Table 3.** The total flavonoid concentration in different ecotypes and one *L. corniculatus* variety under water deficit throughout the seasons.

| Ecotypes/Variety | SWC | Total Flavonoid Concentration (mg QE/gFW) | | | |
|---|---|---|---|---|---|
| | | Summer 2021 | Autumn 2021 | Winter 2021–2022 | Spring 2022 |
| 255301 | 100% FC (NWD) | 16.592 ± 1.4 [a] | 18.440 ± 1.6 [a] | 34.557 ± 6.8 [cd] | 12.898 ± 1.6 [a] |
| | 89% FC (WD) | 19.084 ± 2.1 [a] | 17.478 ± 1.1 [a] | 57.965 ± 0.4 [ab] | 20.454 ± 1.0 [a] |
| 255305 | 100% FC (NWD) | 19.919 ± 1.9 [a] | 19.452 ± 1.4 [a] | 42.826 ± 0.9 [bcd] | 16.385 ± 0.2 [a] |
| | 89% FC (WD) | 17.084 ± 2.5 [a] | 19.253 ± 1.3 [a] | 51.887 ± 3.8 [abc] | 18.975 ± 2.2 [a] |
| 202700 | 100% FC (NWD) | 20.731 ± 0.5 [a] | 17.512 ± 2.8 [a] | 43.294 ± 5.3 [bcd] | 13.975 ± 1.2 [a] |
| | 89% FC (WD) | 16.220 ± 0.5 [a] | 16.517 ± 1.2 [a] | 56.008 ± 3.1 [ab] | 18.703 ± 0.6 [a] |
| 226792 | 100% FC (NWD) | 16.594 ± 0.5 [a] | 15.385 ± 0.8 [a] | 27.187 ± 1.4 [d] | 13.393 ± 1.7 [a] |
| | 89% FC (WD) | 16.415 ± 1.6 [a] | 16.345 ± 2.6 [a] | 53.210 ± 0.4 [ab] | 19.485 ± 0.3 [a] |
| Estanzuela Ganador | 100% FC (NWD) | 19.016 ± 2.3 [a] | 15.070 ± 1.5 [a] | 48.435 ± 3.7 [abc] | 13.076 ± 1.8 [a] |
| | 89% FC (WD) | 16.377 ± 0.2 [a] | 18.839 ± 0.7 [a] | 63.291 ± 4.3 [a] | 19.668 ± 1.8 [a] |

Mean ± standard deviation. Tukey test ($p \leq 0.05$). Figures with the same letter within the same column are statistically equal. SWC = soil water content; QE = Quercetin equivalent; FC = field capacity; NWD = no water deficit; WD = water deficit.

*3.4. Tannins*

The total tannins concentration (TTC) also performed similarly to the TPC and the TFC in the plants evaluated. In winter, there was a significantly greater response ($p \leq 0.05$) in the 255301 and 202700 ecotypes, with 71.7 and 65.3 mg GAE/gFW under a WD. The 202700 ecotype showed the lowest response, with 44.5 mg GAE/gFW under NWD. In addition, the 253301 ecotype presented a significant difference among plant genetic resources in spring 2022, with the highest response of 32.3 mg GAE/gFW under a WD. On the other hand, the 255301 ecotype and the Estanzuela Ganador variety had the lowest response with concentrations of 20.6 and 20.4 mg GAE/gFW under NWD. This means that for these genetic resources, the production of tannins was not required to mitigate the environmental stress associated with low temperatures and the water deficit (Table 4).

**Table 4.** Total tannins concentration in different ecotypes and a *L. corniculatus* variety under water deficit throughout the seasons.

| Ecotypes/Variety | SWC | Total Tannins Concentration (mg GAE/gFW) | | | |
|---|---|---|---|---|---|
| | | Summer 2021 | Autumn 2021 | Winter 2021–2022 | Spring 2022 |
| 255301 | 100% FC (NWD) | 34.331 ± 3.5 [a] | 34.543 ± 3.6 [a] | 48.345 ± 1.3 [bc] | 20.656 ± 3.2 [b] |
| | 89% FC (WD) | 36.300 ± 2.2 [a] | 43.844 ± 3.0 [a] | 71.710 ± 3.4 [a] | 32.353 ± 1.0 [a] |
| 255305 | 100% FC (NWD) | 41.277 ± 3.7 [a] | 37.889 ± 5.6 [a] | 59.382 ± 3.3 [abc] | 23.502 ± 0.7 [ab] |
| | 89% FC (WD) | 38.756 ± 2.6 [a] | 40.720 ± 2.9 [a] | 65.365 ± 4.7 [a] | 27.314 ± 4.1 [ab] |
| 202700 | 100% FC (NWD) | 30.542 ± 5.0 [a] | 33.179 ± 2.7 [a] | 49.949 ± 3.0 [bc] | 22.043 ± 2.7 [ab] |
| | 89% FC (WD) | 32.745 ± 2.3 [a] | 37.277 ± 1.7 [a] | 62.004 ± 2.9 [ab] | 26.880 ± 2.0 [ab] |
| 226792 | 100% FC (NWD) | 36.121 ± 0.7 [a] | 38.794 ± 0.8 [a] | 44.550 ± 1.6 [c] | 23.777 ± 0.7 [ab] |
| | 89% FC (WD) | 30.868 ± 1.8 [a] | 37.666 ± 3.8 [a] | 60.701 ± 3.0 [ab] | 28.751 ± 1.8 [ab] |
| Estanzuela Ganador | 100% FC (NWD) | 41.653 ± 0.5 [a] | 35.806 ± 3.4 [a] | 49.798 ± 3.0 [bc] | 20.471 ± 1.7 [b] |
| | 89% FC (WD) | 26.447 ± 0.1 [a] | 41.340 ± 0.8 [a] | 60.463 ± 1.9 [ab] | 32.006 ± 2.0 [a] |

Mean ± standard deviation. Tukey test ($p \leq 0.05$). Figures with the same letter within the same column are statistically equal. SWC = soil water content; GAE = Gallic Acid Equivalent; FC = field capacity; NWD = no water deficit; WD = water deficit.

### 3.5. Saponins

The total saponins concentration (TSC) had the highest variation throughout the seasons. It had values from 31.2 to 247.3 GAE/gFW in spring and summer, respectively. There was no statistical difference among *Lotus* genetic resources, except in summer 2021 and winter 2021–2022. The 255301 ecotype in a WD was the most heat-sensitive, producing 254.8 mg saponins/gFW, compared to 226792, which showed the lowest concentration with 138.7 mg saponins/gFW (Table 5).

**Table 5.** The total saponins concentration of in different ecotypes and one *L. corniculatus* variety under water deficit throughout the seasons.

| Ecotypes/Variety | SWC | Total Saponins Concentration (mg Saponin/gFW) | | | |
|---|---|---|---|---|---|
| | | Summer 2021 | Autumn 2021 | Winter 2021–2022 | Spring 2022 |
| 255301 | 100% FC (NWD) | 180.460 ± 14.2 [ab] | 161.762 ± 8.9 [a] | 92.652 ± 9.4 [b] | 31.211 ± 3.9 [a] |
| | 89% FC (WD) | 254.869 ± 14.3 [a] | 186.385 ± 1.1 [a] | 145.342 ± 7.0 [a] | 45.301 ± 7.7 [a] |
| 255305 | 100% FC (NWD) | 222.720 ± 11.6 [ab] | 138.888 ± 2.8 [a] | 131.771 ± 3.6 [ab] | 34.169 ± 9.5 [a] |
| | 89% FC (WD) | 247.373 ± 14.6 [ab] | 186.005 ± 2.2 [a] | 107.550 ± 7.6 [ab] | 38.773 ± 8.1 [a] |
| 202700 | 100% FC (NWD) | 176.378 ± 16.5 [ab] | 146.048 ± 1.9 [a] | 140.397 ± 14.3 [ab] | 30.338 ± 6.3 [a] |
| | 89% FC (WD) | 187.382 ± 8.0 [ab] | 171.172 ± 6.9 [a] | 128.749 ± 5.8 [ab] | 42.577 ± 8.4 [a] |
| 226792 | 100% FC (NWD) | 142.177 ± 18.6 [b] | 174.395 ± 2.1 [a] | 122.021 ± 13.2 [ab] | 34.887 ± 7.8 [a] |
| | 89% FC (WD) | 138.890 ± 17.4 [b] | 143.549 ± 9.9 [a] | 120.065 ± 3.9 [a] | 42.595 ± 1.1 [a] |
| Estanzuela Ganador | 100% FC (NWD) | 195.905 ± 3.9 [ab] | 169.869 ± 3.8 [a] | 106.424 ± 7.4 [ab] | 28.424 ± 2.8 [a] |
| | 89% FC (WD) | 183.737 ± 6.9 [ab] | 195.811 ± 3.2 [a] | 136.781 ± 3.4 [ab] | 35.935 ± 2.0 [a] |

Mean ± standard deviation. Tukey test ($p \leq 0.05$). Figures with the same letter within the same column are statistically equal. SWC = soil water content; FC = field capacity; NWD = no water deficit; WD = water deficit.

### 3.6. Radical Scavenging Activity

The DPPH technique detailed the highest radical scavenging activity (RSA). In the 255301 ecotype, the summer, autumn, and winter had values of 78.6, 114.2, and 244.9 mg TEAC/gFW, respectively. In the summer, the 255305 ecotype had a value of 79.8 mg TEAC/gFW, and the 226792 ecotype in winter had a value of 179.4 mg TEAC/gFW, all under a WD. On the other hand, those with the lowest response were the 226792 ecotype in summer, with 27.9 mg TEAC/gFW, and the Estanzuela Ganador variety in autumn and winter (18.7 and 114.7 mg TEAC/gFW) under a WD, but also under NWD with 93.6 mg TEAC/gFW (Table 6).

**Table 6.** Radical scavenging activity in ecotypes and one variety of *L. corniculatus* under water deficit in different seasons.

| Ecotypes/Variety | SWC | Radical Scavenging Activity (mg of TEAC/gFW) | | | |
| --- | --- | --- | --- | --- | --- |
| | | Summer 2021 | Autumn 2021 | Winter 2021–2022 | Spring 2022 |
| 255301 | 100% FC (NWD) | 42.528 ± 9.4 cd | 51.819 ± 5.9 abc | 133.138 ± 12.0 bcd | 47.110 ± 4.8 a |
| | 89% FC (WD) | 78.613 ± 3.6 a | 114.253 ± 10.9 a | 244.910 ± 10.6 a | 70.941 ± 6.7 a |
| 255305 | 100% FC (NWD) | 73.382 ± 3.9 ab | 105.165 ± 10.7 a | 209.139 ± 11.2 abc | 75.290 ± 10.3 a |
| | 89% FC (WD) | 79.895 ± 3.2 a | 87.647 ± 12.8 ab | 221.492 ± 16.3 ab | 64.810 ± 3.1 a |
| 202700 | 100% FC (NWD) | 60.7661 ± 2.2 bc | 85.630 ± 8.2 ab | 165.731 ± 8.9 a–d | 58.623 ± 5.8 a |
| | 89% FC (WD) | 66.822 ± 2.7 abc | 23.665 ± 1.4 bc | 187.358 ± 14.1 a–d | 72.837 ± 2.6 a |
| 226792 | 100% FC (NWD) | 27.992 ± 7.5 d | 24.782 ± 1.1 bc | 141.540 ± 17.0 bcd | 47.339 ± 2.6 a |
| | 89% FC (WD) | 47.137 ± 5.5 bcd | 80.657 ± 4.1 abc | 179.409 ± 8.9 a | 85.027 ± 1.4 a |
| Estanzuela Ganador | 100% FC (NWD) | 43.686 ± 8.9 cd | 62.400 ± 6.6 abc | 93.630 ± 6.1 d | 62.555 ± 7.2 a |
| | 89% FC (WD) | 49.230 ± 3.1 bcd | 15.478 ± 1.6 c | 114.749 ± 9.4 cd | 67.850 ± 10.9 a |

Mean ± standard deviation. Tukey test ($p \leq 0.05$). Figures with the same letter within the same column are statistically equal. SWC = soil water content; TEAC = Trolox Equivalent Antioxidant Capacity; FC = field capacity; NWD = no water deficit; WD = water deficit.

A direct proportional relationship among TAA and CTP ($R^2 = 0.58$), CTF ($R^2 = 0.59$), and CTT ($R^2 = 0.53$) was confirmed (Figure 4a–c) using regression technique, which is congruent, since TAA is an accumulated expression of the different metabolites reactive to the oxidation process due to the effect of environmental stress (Figure 4).

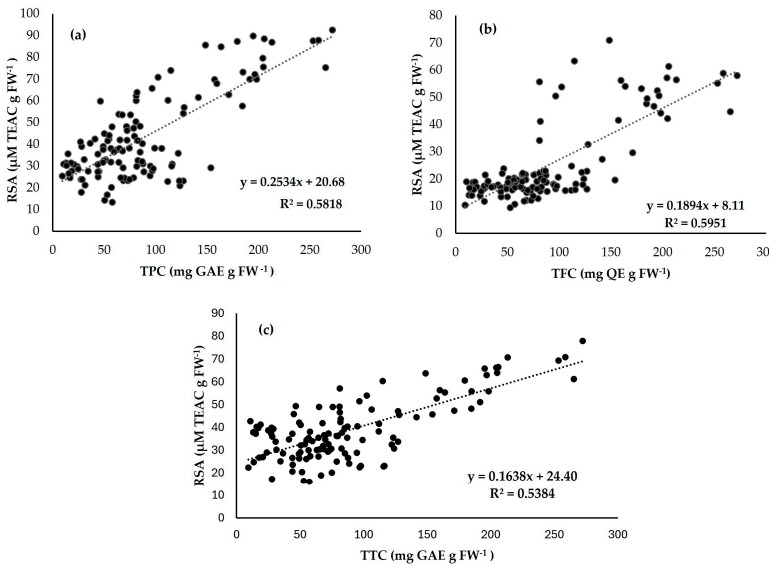

**Figure 4.** Relationship of radical scavenging activity (RSA) with the total phenols concentration (TPC) (**a**), the total flavonoid concentration (TFC) (**b**), and total tannins concentration (TTC) (**c**) in different ecotypes and one *L. corniculatus* variety.

## 4. Discussion

All the evaluated plant genetic resources maintained adequate phenolic compound production in the different seasons. However, the low temperatures in winter increased the output of this secondary metabolite, with the 255301 and 226792 ecotypes being the most susceptible as they produced more phenols to mitigate the cold stress in a WD. Kowalczewski et al. [30] quantified the TPC from 5.00 to 8.16 mg GAE/gFW in *Hordeum vulgare* L. var. KWS Olof, and although the water stress decreased the TPC, the photosynthetically active radiation (PAR) increased. Wong-Paz et al. [16] pointed out that plants in semi-arid areas have high antioxidant capacity due to a high content of total phenols as a defense mechanism against plant stress. Baali et al. [18] reported that in *L. corniculatus* plants

collected in March, the CTP was 87.1 ± 14.5 mg GAE/gFW, and the radical scavenging activity (RSA) was 26.9–79.7 mg TEAC/gFW.

The Estanzuela Ganador variety was susceptible to low winter temperatures, increasing the concentration of flavonoids as a defense mechanism against low temperatures and a water deficit. The 226792 ecotype was the most tolerant to winter conditions, with the most deficient production of this metabolite under NWD and without requiring the production of this metabolite to mitigate the WD. The TFC increased in response to low temperatures and a water deficit, with values higher than those -reported by Baali et al. (2019) [18], who documented values of 36.5 ± 2.1 mg QE/gFW in TFC in *L. corniculatus* extracts in response to environmental stress intensity. Additionally, the TFC in *L. corniculatus* varies depending on the environmental temperature; for example, it increases at 10 °C and decreases at 30 °C [31]. The decrease in TFC in summer, autumn, and spring could be due to the natural tolerance of most plant genetic resources evaluated in this study to the extreme temperatures in these seasons, as they do not require the antioxidant activity that promotes flavonoids. Cheng et al. [32] reported that only low temperatures, and not high temperatures, encourage the expression of genes fitted to produce flavonoids and, therefore, their accumulation.

The response of the 255301 and 255792 ecotypes and the Estanzuela Ganador variety had high tannin concentrations in winter and spring, except for the 255792 ecotype, which only reacted this way in winter. All the mentioned responses were under a water deficit. This secondary metabolite signifies the importance of mitigating the abiotic stress associated with low temperatures and a water deficit. Previous studies in three *Lotus* species (*L. glaber*, *L. uliginosus*, and *L. corniculatus*) found a high tannin content in the leaves, stems, and flowers, which varied in the different seasons of the year, except for *L. glaber*, for which there were no differences in spring, summer, and autumn [33,34]. The increase in tannins observed in winter could be due to the maturity of the plants and the increase in lignin, thereby decreasing the forage digestibility [35].

Saponins were the most immediate secondary metabolite to the elevated temperatures reported in summer. At the same time, the TPC, TFC, and TTC were more sensitive to low temperatures in winter and high temperatures in summer under a water deficit, with different responses among the plant genetic resources of *Lotus*. Szakiel et al. [36] described that saponins could be involved in adapting plants to survive in adverse climatic conditions, with a positive correlation between the saponin content and the physiological phase throughout the year. Baali et al. [18] detected the presence of saponins in *L. corniculatus* using the qualitative foam test. Santacoloma [37] reported the absence of saponins in the leaves, petioles, and stems of *L. corniculatus*, which may be associated with the favorable environmental conditions of the study area.

The antioxidant activity results were correlated to the production of some secondary metabolites in the content of phenols, flavonoids, and tannins in winter under a water deficit, particularly in the 255301 and 255305 ecotypes, as well as the Estanzuela Ganador variety, but only in terms of flavonoid production. On the other hand, saponins did not contribute to the antioxidant activity in the winter phase. Still, they did contribute in the summer, which is the time of highest temperatures, in the 255301 and 266792 ecotypes, which were under a water deficit since they were the phylogenetic materials with the highest RSA in that season. This suggests that saponin production is activated as a defense mechanism against heat and a water deficit. Indeed, antioxidant compounds are produced as a defense mechanism to prevent growth inhibition and damage to the photosynthetic apparatus, cell membranes, and proteins, with a specific response to a water deficit dependent on the genotypes used [38].

Finally, it is essential to point out that this study was under semi-controlled shade mesh conditions, so the results presented here require validation in the field.

## 5. Conclusions

The 255301 ecotype increased the concentration of phenols, flavonoids, and tannins under a water deficit. In contrast, the 255305 ecotype improved the tannin concentration and total antioxidant activity under both no water deficit and water deficit conditions, respectively, in winter when the temperatures were lower. The 202700 and 226792 ecotypes and the Estanzuela Ganador variety showed similar performance in soil moisture content, with lower values in secondary metabolite concentrations and antioxidant activity throughout the year. The 255301 ecotype showed an increase in the concentration of saponins as a mitigation response to stress due to the high temperatures and water deficit in summer. This diversity of responses indicates a high potential for *L. corniculatus* as an alternative forage crop in arid lands.

**Author Contributions:** Conceptualization, A.P.-S., L.A.G.-E. and G.G.d.l.S.; methodology, L.A.G.-E., A.P.-S., R.T.-C. and P.Á.-V.; software, L.A.G.-E., A.P.-S. and R.T.-C.; validation, L.A.G.-E., A.P.-S. and M.d.R.J.-S.; formal analysis, L.A.G.-E. and A.P.-S.; investigation, A.P.-S., L.A.G.-E., R.T.-C. and M.d.R.J.-S.; resources, A.P.-S. and L.A.G.-E.; data curation, A.P.-S., G.G.d.l.S. and L.A.G.-E.; writing—original draft preparation, L.A.G.-E., A.P.-S. and P.Á.-V.; writing—review and editing, A.P.-S., L.A.G.-E. and R.T.-C.; project administration, A.P.-S. and R.T.-C. All authors have read and agreed to the published version of the manuscript.

**Funding:** This research was supported by the Dirección General de Investigación y Posgrado—Universidad Autónoma Chapingo through the project ID 21006-EI.

**Institutional Review Board Statement:** Not applicable.

**Informed Consent Statement:** Not applicable.

**Data Availability Statement:** Data are available on request to the corresponding author's email with appropriate justification.

**Acknowledgments:** Thanks to the Institute of Natural Resources of the Postgraduate College at Montecillo, Mexico, for the donation of the plant genetic resources of *Lotus corniculatus* L.

**Conflicts of Interest:** The authors declare no conflicts of interest.

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
