# Peer review of "Secondary Metabolites and Their Antioxidant Activity Enhance the Tolerance to Water Deficit on Clover Lotus corniculatus L. through Different Seasonal Times"

_2037-0164, doi:10.3390/ijpb15010014_

Round 1

Reviewer 1 Report

All abbreviation should be embedded at the first mention. 

In the Introduction section, please, bring light into the role of secondary metabolites to tolerate water deficite. 

What is the reason for the accessions selection of Lotus corniculatus?

Taking in account that 5 Lotus origins were studied, is the study sufficiently reliable?

DPPH is a method for free radical scavendging. I suggest additional  assays of antioxidant activity to be performed. 

All abbreviation should be embedded at the first mention. 

In the Introduction section, please, bring light into the role of secondary metabolites to tolerate water deficite. 

What is the reason for the accessions selection of Lotus corniculatus?

Some details from the first paragraph of Discussion section should be moved to the methods (l.296-306).

Taking in account that 5 Lotus origins were studied, is the study sufficiently reliable?

DPPH is a method for free radical scavendging. I suggest additional  assays of antioxidant activity to be performed. 

Check for typos errors.

Reviewer 2 Report

This article studied the effects of water limitation in different ecotypes and one 16 variety of Lotus corniculatus L. on the production of secondary metabolites.Overall, the results are clear and unambiguous.However, there are some minor problems that need to be modified

first, the introduction is too scattered, which can be considered modified.Second, there are some formatting problems such as the Spaces in part 3.2, and R2 in Figure 2 is recommended to keep four decimal places. In addition, the recommended format for the data in the tables is mean with sd.

Round 2

Reviewer 1 Report

The Authors have addressed my comments from the first round. The manuscript has been improved according to all my suggestions. I have no more remarks.

The Authors have addressed my comments from the first round. The manuscript has been improved according to all my suggestions. I have no more remarks.
